# Metabolic response to an acute bout of mild dynamic exercise performed under normobaric moderate hypoxia: A NMR-based metabolomics study

Flaminia Cesare Marincola[1]*, Daniela Masu[1], Veronica Libonati[1], Michela Tozzi[1], Raffaella Isola[2], Romina Vargiu[2], Elisabetta Marini[3], Silvana Roberto[4], Sara Magnani[4], Giovanna Ghiani[4], Gabriele Mulliri[4]*, Antonio Crisafulli[4], Andrea C. Rinaldi[2]

1 Department of Chemical and Geological Sciences, University of Cagliari & CSGI, Monserrato, Cagliari, Italy, 2 Department of Biomedical Sciences, University of Cagliari, Monserrato, Cagliari, Italy, 3 Department of Life and Environmental Sciences, Section of Neuroscience and Anthropology, University of Cagliari, Monserrato, Cagliari, Italy, 4 Department of Medical Sciences and Public Health, University of Cagliari, Monserrato, Cagliari, Italy.

* flaminia@unica.it (FCM); jabutele84@gmail.com (GM)

## Abstract

The combination of hypoxia and exercise offers significant potential benefits for non-athletic individuals, particularly in clinical and rehabilitation settings. Personalized hypoxic exercise programs can be tailored to improve cardiovascular and metabolic health, enhance recovery, and promote adaptation to low-oxygen environments. However, finding the optimal balance between exercise variables (intensity, duration, frequency, type of exercise), and hypoxic exposure parameters (altitude level, duration, session frequency), remains challenging. Further research is needed to understand how these variables interact to optimize hypoxic exercise protocols. In the present study, we explored the effects of a single session of mild dynamic exercise conducted in normobaric hypoxia (FiO$_2$=13.5%) on the plasma and urine metabolome of thirteen healthy young adults (age 29.7±4.5 y, body mass index 23.5±1.4 kg/m$^2$). For comparative purposes, participants performed the same exercise under normoxia (FiO$_2$=21%). During both exercise sessions, subjects wore a mask connected to a hypoxic gas generator while seated on a cycle ergometer. After a 4 minute rest, they pedaled for 3 minutes at 30% of their Wmax, followed by 6 minutes of recovery. Hemodynamic parameters were measured at four time points, and biological samples (blood and urine) were collected before the test and within 5 minutes of exercise completion. Samples were analyzed by $^1$H NMR spectroscopy. Univariate and multivariate statistical analysis of NMR datasets revealed noteworthy changes in the levels of certain metabolites following the hypoxic session: 3-hydroxybutyrate, branched-chain amino acids, citrate, lactate, phenylalanine, succinate, and tyrosine in plasma; 3-hydroxyisobutyrate, 3-hydroxyisovalerate, alanine, acetone, dimethylamine, glycine, lactate, succinate, and taurine in urine. These metabolic shifts, along with their

**Data availability statement:** The datasets supporting the findings of this study have been deposited in the Zenodo public repository (https://doi.org/10.5281/zenodo.15228863).

**Funding:** This research was funded by Regione Autonoma della Sardegna (RAS) FSC 2014-2020, call on basic research 2017 (Grant SR 85013), PI Antonio Crisafulli (https://www.regione.sardegna.it) and by a PRIN PNRR research grant from the Italian Ministry of Research and University, awarded to Andrea C. Rinaldi, within the project entitled "Tapping into the biological potential of wild mushrooms from a range of ecosystems in Sardinia and Abruzzi: FUNSarAbr (F53D23012210001). The funder had no role in study design, data collection and analysis, decision to publish, or preparation of the manuscript.

**Competing interests:** The authors have declared that no competing interests exist.

statistically significant correlations with hemodynamic parameters, suggest an adaptive modulation of energy metabolism pathways in response to mild hypoxic stress.

## Introduction

At high altitudes or in simulated low-oxygen settings, the reduced oxygen partial pressure leads to tissue hypoxia, challenging the body's physiological systems. While adaptations to hypoxic conditions have traditionally been emphasized for elite athletes, due to potentially enhance exercise performance [1], they also offer significant benefits for non-competitive individuals, including sedentary and clinical populations. For these groups, passive (i.e., during rest) or active (i.e., during exercise) exposure to hypoxic stimuli can assist in managing health issues such as hypertension, cardiovascular diseases, and obesity [2–4]. Additionally, hypoxic exposure can aid in rehabilitation by speeding up recovery and improving functional outcomes [5], and it can serve as an effective strategy for pre-acclimatization to activities like mountaineering or environments with low oxygen levels [6,7]. Given practical constraints such as work and family obligations, non-athletic populations, particularly clinical groups, often lack access to natural high-altitude environments. As a result, these groups commonly turn to simulated altitude techniques, adapted from the well-established hypoxic training methods used by elite athletes to mimic hypobaric or normobaric hypoxic environments. Hypobaric hypoxia recreates high-altitude conditions by lowering ambient pressure, mimicking the physiological stress caused by reduced oxygen levels at altitude. Differently, normobaric hypoxia maintains normal barometric pressure but reduces the oxygen concentration in the air, offering a more accessible and practical way to expose individuals to low oxygen without the added strain of decreased air pressure. Gaining insight into the metabolic adaptations of non-athletic individuals to hypoxic conditions is crucial for developing effective rehabilitation and health strategies. By targeting these adaptations, it becomes possible to tailor accessible and adaptable hypoxic programs to improve recovery and optimize health outcomes in non-athletic populations.

Metabolomics, the study of metabolome, i.e., the pool of low molecular weight (<1.5 kDa) metabolites in a biological sample such as biofluids, tissues, cells, and organs, has been shown to be a powerful tool for studying hypoxia-induced metabolic shifts [8–10]. Indeed, since these metabolites represent the intermediates or end-products of the gene expression, this approach can give an instantaneous snapshot of the cell functional status of an individual, allowing researchers to capture the dynamic changes occurring in the body under stress [11]. To the best of our knowledge, the application of metabolomics to study the impact of hypoxic conditions under exercise is still in its early stages. To date, only three studies have investigated how hypoxia affects the human metabolome during physical activity, with two focusing on trained athletes [12,13] and one on recreational sportsmen [14].

Our research group has concentrated its recent efforts on studying the effects of acute (lasting only a few minutes) normobaric hypoxia on the cardiovascular and

metabolic responses of both athletic and non-athletic individuals, particularly during low-intensity exercise [15–17]. This research aims to fill gaps in understanding how brief hypoxic exposure influences key physiological functions across diverse populations. In particular, in a recent study, we have analyzed the hemodynamic responses of healthy young adults during a session of mild dynamic exercise under moderate normobaric hypoxia. [18]. Our findings revealed a significant decrease in peripheral arterial oxygen saturation and cerebral oxygenation during hypoxic activity compared to exercise performed under normal oxygen conditions. Nevertheless, we observed that the circulatory system was able to handle this hypoxic condition well. In the current investigation, we extended our previous research to assess at the molecular level the short-term effects of the above-mentioned mild exercise performed under moderate normobaric hypoxia on the human metabolome of thirteen healthy young adults engaged in leisure-time physical activities. In order to achieve this, we used an NMR-based metabolomics approach to analyze two biofluids, i.e., blood and urine, collected both before and just after the exercise. Participants also underwent an identical exercise session in normoxia for comparison's purposes.

## Materials and methods

### Study design

This study was designed as a crossover randomized controlled trial, in which each participant completed two experimental conditions. Participants visited the laboratory on three separate occasions. They underwent a preliminary screening test, and then, following an interval of 4–7 days, two constant-load exercise tests with a 7–10 day gap between them to minimize any carryover effects. The tests differed in the fraction of inspired oxygen ($FiO_2$): the NORMO test was performed in condition of normobaric normoxia ($FiO_2$ 21%), while the HYPO test under normobaric hypoxia ($FiO_2$ 13.5%, corresponding to an altitude of approximately 3500 m). The level of hypoxia was chosen considering previous studies on similar experimental setting [16–21]. The assignment of NORMO and HYPO tests was randomized using an online random sequence generator (https://www.random.org/sequences/) to prevent any order effect. During each experimental session, hemodynamic parameters were monitored, and urine and blood samples were collected before and after exercise.

The required sample size was calculated using an online sample size tool (Statulator, https://statulator.com/SampleSize/ss2M.html) based on the following parameters: 1) 85% statistical power; 2) a type 1 error rate (α) of 0.05; 3) a standard deviation (SD) of 20%; 4) an expected 25% difference between conditions in the studied variables. This analysis indicated that a minimum of 11 participants was required to detect significant differences. To account for potential dropouts, we recruited 15 subjects. There were no deviations from the original study design, with the exception that two participants were excluded post hoc due to suspected non-compliance with dietary restrictions, as identified through [1]H NMR analysis of urine which revealed unusually high levels of erythritol, a sugar alcohol commonly used as a low-calorie sweetener in various processed foods and beverages. Consequently, the final cohort for analysis included 13 participants. A schematic overview of the study design, including the randomization procedure, is provided in Fig. 1.

The study was conducted according to the Declaration of Helsinki and was approved by the ethics committee of the University of Cagliari (letter n. 0073832–30/03/2021). All the participants signed written informed consent before the beginning of the study.

### Participants

The participants recruitment took place from April 2021 to October 2022 through voluntary enrollment via advertisements in various sports clubs. To be eligible, participants had to be healthy, non-smoking men aged 20–40 years, and involved in leisure-time physical activities at least three times/week. Exclusion criteria included any history of cardiovascular, metabolic, or respiratory diseases, as well as prior exposed to altitude above 1000 m over the past three months.

All participants underwent a preliminary medical examination to assess their health status. None of them were on medication at the time of the experiment. All subjects were instructed to avoid alcohol and intense exercise for at least 48

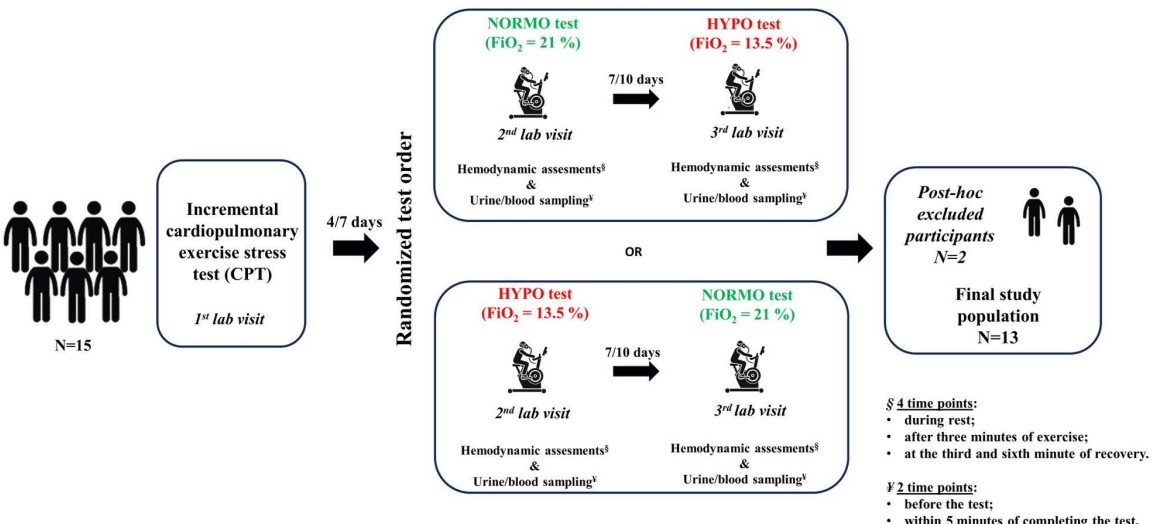

**Fig 1. Overview of the experimental design.** Fifteen participants attended the laboratory on three separate occasions. During the first visit, they underwent an incremental cardiopulmonary exercise stress test (CPT). In the following two sessions, spaced 7 to 10 days apart, they completed two constant-load exercise tests: one under normoxia (NORMO) and the other under normobaric hypoxia (HYPO). The order of the two sessions was randomized to prevent order effects. Hemodynamic parameters were assessed at four time points, and urine and blood sampling were done before and after each exercise session. Out of the 15 initially recruited participants, two were excluded post hoc due to suspected non-compliance with dietary restrictions, resulting in a final analyzed sample of 13 participants.

hours prior to the session, to abstain from caffeinated drinks the day of the testing and were advised to sleep six to eight hours the night before the tests.

## Screening test

The screening test began with a preliminary medical examination of subjects to exclude any cardiovascular or respiratory diseases. Following the medical assessment, participants underwent an incremental cardiopulmonary exercise stress test (CPT) while their heart rate (HR) and electrocardiogram (ECG) were recorded. The CPT was performed on a mechanically-braked cycle ergometer (Monark 828E, Vansbro, Sweden). It entailed a gradual increase in workload (30 W/min), commencing at 30 W, with a pedaling rate of 60 rpm, continuing until exhaustion, defined as the point at which the subject couldn't maintain a pedaling rate of at least 50 rpm. The maximum workload ($W_{max}$) and maximum oxygen uptake ($VO_{2max}$) were measured during the test. Achievement of $VO_{2max}$ was determined by meeting at least two of the following criteria: (1) a plateau in oxygen uptake ($VO_2$) despite increasing workload ($< 80$ mL·min$^{-1}$), (2) a respiratory exchange ratio exceeding 1.10, and (3) HR within $\pm 10$ beats·min$^{-1}$ of the predicted maximum HR, calculated as 220-age. $VO_{2max}$ was calculated as the average $VO_2$ during the final 30 seconds of the incremental test. $VO_2$, carbon dioxide production ($VCO_2$), and ventilation (VE) were measured using a gas analyzer (ULTIMA CPX, MedGraphics, St. Paul, MN, USA) and calibrated immediately before each test, following the manufacturer's guidelines. During the preliminary test, participants familiarized themselves with the laboratory personnel and equipment, facilitating adaptation to the environment and the cycle ergometer utilized in subsequent experimental sessions.

## Experimental trials

The NORMO and HYPO tests were performed in the morning, at the same time of the day (ca 2 h 30 min after consumption of a light breakfast) to reduce circadian bias. In each test, subjects seated on a cycle ergometer. After a 4 minute

rest, they pedaled for 3 minutes at 30% of their Wmax, followed by 6 minutes of recovery. During both sessions, subjects fitted to a face mask connected to a hypoxic gas generator (EverestSummit II Generator, Hypoxico, New York, USA). This device separates nitrogen from oxygen thanks to a molecular sieve system that uses zeolites and provides a gas mixture with a reduced oxygen content that can be regulated to reach a minimum $FiO_2$ of 12.5%. Throughout the tests, $FiO_2$ was constantly checked by an operator using an oxygen analyzer provided with the device (Maxtec, Handi+, Salt Lake City, UT, USA). Participants were blinded about the content of oxygen they were breathing.

### Hemodynamic assessment

The assessment of hemodynamic parameters was done as previously described [18]. Briefly, hemodynamic data were collected by using the technique of impedance cardiography, measuring changes in thoracic impedance to calculate stroke volume (SV). Offline analysis included calculating heart rate (HR). Cardiac output (CO) was calculated as the product of SV and HR. Systolic (SBP) and diastolic blood pressure (DBP) were measured by applying manually a sphygmomanometer on the non-dominant arm. To avoid potential operator-dependent biases, these measurements were conducted by the same physician. Mean arterial pressure (MAP) was determined from SBP and DBP using a formula that adjusts for changes in diastolic time and systolic time during exercise-induced tachycardia [22]. Systemic vascular resistance (SVR) was computed by dividing MAP by CO and multiplying the result by 80, where 80 serves as a conversion factor to achieve standard resistance units.

### Biological sampling

In each exercise session, urine and blood samples were gathered at two specific time points: before the test and within 5 minutes of completing the test. Urine was collected in an aseptic container, transferred into centrifuge tubes and centrifuged at $1800 \times g$ for 10 min at 4°C to remove solid particles. Blood was taken by venipuncture from an antecubital vein of the forearm using heparin vacutainers. Plasma was obtained by immediately centrifuging tubes at $1600 \times g$ for 15 min at 4°C. Both urine and plasma were aliquoted into 2 ml cryovials and frozen at −80 °C until analysis.

### Sample preparation for NMR analysis

On the day of NMR analysis, biofluids were thawed in ice. Plasma samples were transferred to 0.5 mL 10K Amicon Ultra centrifugal filter units (Merck Millipore Ltd., Cork, Ireland) and centrifuged at $13000 \times g$ for 40 min at 4 °C. Prior to filtration, the filters were washed out from glycerol by adding 500 µl of distilled water and then, centrifuging at $13000 \times g$ for 10 min at room temperature. The procedure was carried out until control of the wash water by [1]H NMR spectroscopy showed no residual presence of glycerol. Then, 300 µl of each filtered plasma sample were diluted with 400 µl of a 0.1 M phosphate buffer solution in $D_2O$ (99.9 atom % D, Sigma-Aldrich, Milano, Italy) at pH = 7.4 containing the internal standard sodium salt of 3-(trimethylsilyl) propionic-2,2,3,3-$d_4$ acid (TSP, 98 atom % D, ≥ 98.0%, Sigma-Aldrich, Milano, Italy) at a 0.21 mM final concentration. An aliquot of 650 µL of the final sample was transferred into a 5 mm NMR tube.

A 900 µL aliquot of urine sample was transferred into a 2 mL vial and mixed with 9 µL of 10% aqueous sodium azide ($NaN_3$, ≥ 99.5%, Sigma-Aldrich, Milano, Italy) solution, used as an antibacterial, and 100 µL of phosphate buffer solution in $D_2O$ (1.5 M, pH 7.4) containing the internal standard TSP at a 1 mM final concentration. Samples were centrifuged at $12000 \times g$ for 10 min at 4°C. Then, 900 µL of supernatant were taken and the pH of the solution was corrected to 7.00 ± 0.05 by adding sodium deuteroxide (40 wt % in $D_2O$, 99.5 atom % D, Sigma-Aldrich, Milano, Italy) and deuterium chloride (35 wt % in $D_2O$, ≥ 99 atom % D, Sigma-Aldrich, Milano, Italy). Finally, 650 µL of the solution were transferred into a 5 mm NMR tube.

### [1]H NMR data acquisition and processing

[1]H NMR spectra were acquired at 300 K using a Varian Unity Inova 500 NMR spectrometer (Agilent Technologies, Santa Clara, CA, USA). A standard pulse sequence, 1D NOESY (noesypresat), with presaturation during relaxation and mixing

time for water suppression was used. NMR experiments were performed using an acquisition time of 1.5 s, 32 K data points, 256 scans over a spectral width of 6000 Hz, a 3.5 s relaxation delay and 1 ms mixing time.

NMR spectra were processed by using MestReNova (Version 14.0, Mestrelab Research SL, Valencia, Spain). After Fourier transformation with an exponential weighting factor corresponding to a line broadening of 0.3 Hz, phase and baseline corrections were manually performed. The chemical shift scale was set by assigning a value of $\delta = 0.00$ ppm to the signal of the internal standard TSP. The assignment of $^1$H NMR signals was performed using the Chenomx NMR Suite software (version 8.2, Edmonton, Canada, evaluate version), by considering published literature data [23] and the Human Metabolome Database (HMDB, available at http://hmdb.ca) [24]. After correction of spectra for misalignments in chemical shift primarily due to pH-dependent signals, the upfield region containing TSP signal (−0.5–0.5 ppm) and the residual water (4.6–5.2 ppm) region were excluded. Additionally, in urine spectra, the region containing urea peak (5.5–6.1 ppm) was removed, whereas in plasma, the lactate doublet (1.33 ppm) was left out due to its significantly higher intensity compared to other peaks. Then, all spectra were integrated into bins with a small equidistant width (0.002 ppm) and normalized using the Probabilistic Quotient Normalization (PQN) method [25].

## Statistical analysis

The analysis of NMR datasets was performed by using both multivariate and univariate statistical methods. For multivariate analysis, data matrices were inputted into SIMCA 17 (Sartorius Stedim Data Analytics AB). Following Pareto scaling, principal component analysis (PCA) was first performed to detect systemic variations, intrinsic clusters, and outliers. A supervised orthogonal projections to latent structure–discriminant analysis (OPLS–DA) was done to optimize the separation between groups and enhance the understanding of the variables responsible for discrimination. The quality of the OPLS-DA models was described by the values of the cumulative modeled variation in the X matrix ($R^2X$), the cumulative modeled variation in the Y matrix ($R^2Y$), and the cross-validated predictive ability ($Q^2$). Evaluation of the reliability of the models was done by default internal SIMCA seven-fold cross-validation (CV-ANOVA) which provides p-value indicating the level of significance for group separation (p-value < 0.05) [26]. Furthermore, a permutation test (400 permutations) was done to validate the predictive capability of the computed models, thus discarding overfitting [27]. Statistically significant variables contributing to class separation were selected by analyzing the coefficient color coded loading plots that depicts the modeled covariance p(cov) and correlation p(corr) coefficients between a variable and the classification score. Potential biomarkers were selected using a significance level |p(cov)| ≥ 0.05 and |p(corr)| ≥ 0.5.

Two-way-ANOVA with repeated measures was employed to examine the effect of oxygen exposure, exercise, and their interaction on cardiovascular parameters and the relative content of identified metabolites by using the peak area of corresponding signals. Analyses were performed using Jamovi statistical software, version 2.3.26 (https://www.jamovi.org/). Before analysis, the normal distribution of variables was evaluated with the Shapiro–Wilk test. Where the data were not normally distributed, a log transformation was applied to meet the normality assumption. In addition, z-scores were calculated to standardize the data and identify potential outliers; no outliers were detected. Bonferroni post hoc test was used for pairwise comparisons when ANOVA revealed a significant main effect or interaction effect (p-value < 0.05). The partial eta-squared ($\eta^2$) was reported as a measure of effect size (small ≥ 0.01 to < 0.06, medium ≥ 0.06 to < 0.13, large ≥ 0.14). A post-hoc power analysis was performed using G*Power software version 3.1.9.7. Pearson and Sperman coefficients were calculated for correlation analysis.

## Results

### Cardiovascular parameters

The current study involved thirteen participants who regularly engaged in leisure-time physical activities at least three times a week. Table 1 shows the baseline characteristics and physical performance of the study population recorded during CPT. Subjects were all males with an average age of 29.7 ± 4.5 y and an average BMI of 23.5 ± 1.4 kg/m$^2$. The

**Table 1. Anthropometric, resting and exercise cardiorespiratory values of study population.**

| | | Value (N = 13) |
|---|---|---|
| **General characteristics** | | |
| Age (yr) | | 29.7 ± 4.5 |
| Weight (Kg) | | 74.1 ± 6.7 |
| Height (cm) | | 177.2 ± 6.1 |
| BMI (kg/m²) | | 23.5 ± 1.4 |
| BSA (m²) | | 1.9 ± 0.1 |
| Resting SBP (mm Hg) | | 114.6 ± 15.9 |
| Resting DBP (mm Hg) | | 69.5 ± 11.7 |
| Resting HR (min$^{-1}$) | | 84.8 ± 12.7 |
| **CPT data** | | |
| | $W_{max}$ | VT1 |
| Workload (W) | 255.4 ± 42.6 | 138.7 ± 31.5 |
| $VO_2$ (ml/kg/min) | 36.1 ± 5.2 | 20.5 ± 4.9 |
| HR (min$^{-1}$) | 177.4 ± 10.2 | 141.7 ± 11.2 |
| $VCO_2$ (ml/min) | 3586.8 ± 40.9 | 1615.0 ± 399.9 |
| VE (l/min) | 98.9 ± 12.8 | 35.9 ± 9.4 |

Values are means ± standard deviation (SD).

BMI, body mass index; BSA, body surface area; DBP, diastolic blood pressure; HR, heart rate; SBP, systolic blood pressure; $VCO_2$, carbon dioxide production; VE, pulmonary ventilation; $VO_2$, maximal oxygen uptake; VT1, first ventilatory threshold; $W_{max}$, maximal power.

table includes measurements taken at the maximal workload (Wmax) and at the first ventilatory threshold (VT1). The average heart rate (HR) at VT1 was 141.7 ± 11.2 bpm, and the power output 138.7 ± 31.5 watts. The oxygen consumption ($VO_2$) averaged 20.5 ± 4.9 ml/kg/min, while the carbon dioxide production ($VCO_2$) and the minute ventilation (VE) were 1615 ± 399.9 ml/min and 35.9 ± 9.4 l/min, respectively. During the maximal exertion phase of the test ($W_{max}$ = 255.4 ± 42.6 W), the $VO_2$ recorded was 36.1 ± 5.2 ml/kg/min, indicating that the subjects had an aerobic fitness level typical of recreationally active individuals. The average $VCO_2$ was 3586.8 ± 40.9 ml/min.

In line with our previous study [18], the two-way analysis of variance (two-way ANOVA) on cardiovascular parameters recorded during the NORMO and HYPO tests revealed that the circulatory system of subjects effectively tolerated well hypoxic exercise (Table 2). Specifically, we found no significant difference in heart rate (HR) between the two tests, indicating unchanged pumping capacity of the heart under hypoxic conditions. Stroke volume (SV) remained consistent during the HYPO exercise, suggesting the lack of substantial cardiovascular stimulus. Our findings also evidenced the absence of significant vasodilation in response to the HYPO session, as systemic vascular resistance (SVR) did not differ between the two tests. This suggested that the exercise was either too brief or too mild to induce metabolite production capable of triggering significant vasodilation. Additionally, during the exercise phase of the NORMO and HYPO tests, both CO and MAP increased without any detectable effect of condition.

## Metabolomics analysis

A total of 104 samples (52 from the NORMO test and 52 from the HYPO test) were collected and analysed by $^1$H NMR spectroscopy. For each biofluid, specimens were categorized into four groups: PRE-N and PRE-H (before the NORMO and HYPO sessions, respectively); POST-N and POST-H (after the NORMO and HYPO sessions, respectively). Initially, PCA was applied to obtain a global overview of the two NMR datasets. Fig 2 shows the PC1 vs PC2 scores plots of the plasma (Fig 2A) and urine (Fig 2B) models, which account for approximately 45 and 26% of the total variance,

**Table 2. Hemodynamic parameters during rest (Pre), after three minutes of exercise (Exe3), and at the third and sixth minute of recovery (Rec 3 and Rec6, respectively) of the tests in normoxia (NORMO) and in normobaric hypoxia (HYPO).**

| | HR (bpm) | |
| --- | --- | --- |
| | NORMO | HYPO |
| Pre | 84.0 ± 13.1 | 80.9 ± 13.0 |
| Exe3 | 123.2* ± 18.1 | 129.8* ± 16.6 |
| Rec3 | 84.4 ± 16.6 | 84.2 ± 18.8 |
| Rec6 | 85.4 ± 13.1 | 84.4 ± 14.3 |
| | CO (L/min) | |
| | NORMO | HYPO |
| Pre | 5.0 ± 1.4 | 4.3 ± 1.2 |
| Exe3 | 12.0* ± 3.2 | 12.7* ± 3.0 |
| Rec3 | 5.3 ± 1.9 | 5.4 ± 1.9 |
| Rec6 | 4.9 ± 1.8 | 5.0 ± 1.6 |
| | MAP (mmHg) | |
| | NORMO | HYPO |
| Pre | 78.9 ± 8.4 | 81.4 ± 9.6 |
| Exe3 | 97.4* ± 12.5 | 99.0* ± 7.7 |
| Rec3 | 83.8 ± 7.0 | 83.0 ± 8.8 |
| Rec6 | 80.2 ± 8.7 | 80.7 ± 8.1 |
| | SV (ml) | |
| | NORMO | HYPO |
| Pre | 60.0 ± 17.9 | 55.2 ± 20.7 |
| Exe3 | 98.5* ± 30.1 | 98.7* ± 24.3 |
| Rec3 | 64.0 ± 22.8 | 68.8 ± 36.4 |
| Rec6 | 58.6 ± 24.0 | 61.3 ± 26.1 |
| | SVR (dynes· sec/cm$^5$) | |
| | NORMO | HYPO |
| Pre | 1374.0 ± 462.2 | 1600.2 ± 438.4 |
| Exe3 | 690.1* ± 178.9 | 651.9* ± 164.5 |
| Rec3 | 1416.9 ± 545.4 | 1337.4 ± 403.7 |
| Rec6 | 1430.0 ± 414.5 | 1419.1 ± 460.2 |

HR, Heart rate; CO, cardiac output; MAP, mean arterial pressure; SV, stroke volume; SVR, systemic vascular resistance. Data are expressed as mean ± SD. * = $p < 0.05$ vs. rest period of the same test.

respectively. Neither of the two plots showed clustering according to the sampling time (PRE- *vs.* POST-exercise) and/or the type of test (NORMO *vs.* HYPO). Furthermore, some outliers were detected in both plots. However, they were not removed from the subsequent statistical analyses. This decision was made because their corresponding NMR spectra did not exhibit any abnormalities that would warrant their exclusion from the datasets.

A further examination of the plasma and urine datasets was conducted through a discriminant multivariate analysis. OPLS-DA models were built for a pair-wise comparison between pre- and post-exercise groups. The average performance statistics of the models are reported in Table 3.

The OPLS-DA models built with samples collected before and after the NORMO test (i.e., PRE-N *vs.* POST-N) were not significant ($Q^2 < 0$). Differently, when considering the PRE-H *vs.* POST-H comparison, a good distinction among the metabolic characteristics of both plasma and urine spectra groups was obtained. The corresponding scores plots are depicted in Fig 3.

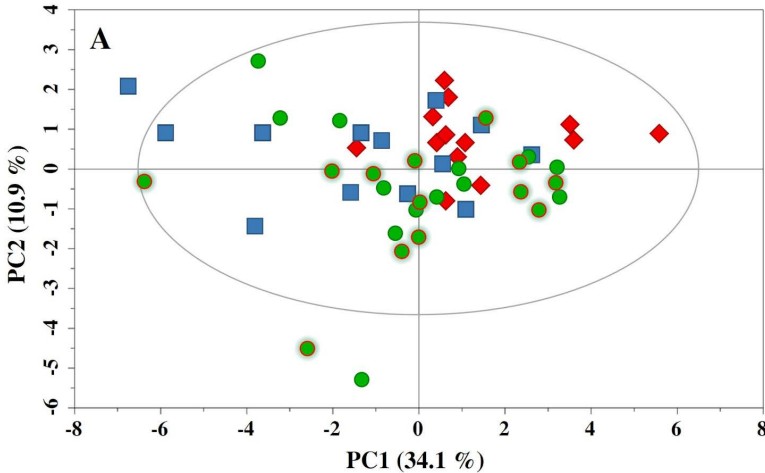

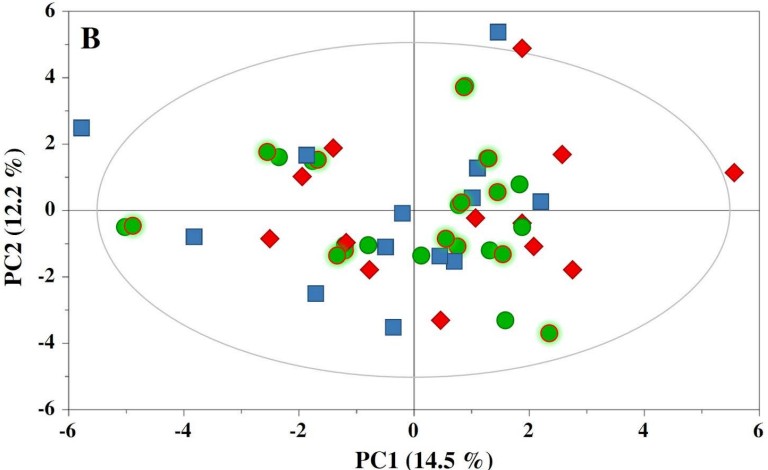

**Fig 2. Principal component analysis (PCA) of plasma and urine NMR datasets.** PCA of plasma and urine samples didn't show any clustering according to the sampling time and/or oxygen exposure. **(A)** PC1 vs PC2 scores plot based on the $^1$H NMR spectra of plasma. **(B)** PC1 vs PC2 scores plot based on the $^1$H NMR spectra of urine. Symbols: •, before test (circle with green border: PRE-N; circles with red border: PRE-H); ■, after NORMO session; ♦, after HYPO session.

**Table 3. Statistical parameters for the OPLS-DA models built with the $^1$H NMR spectra of plasma and urine for the pairwise comparison PRE-N** *vs.* **POST-N and PRE-H** *vs.* **POST-H[a].**

| Pairwise comparison | | | | Permutation test (n = 400) | | |
|---|---|---|---|---|---|---|
| | R²X | R²Y | Q²Y | R²Y intercept | Q²Y intercept | p-value |
| *Plasma* | | | | | | |
| PRE-N *vs.* POST-N | 0.343 | 0.186 | −0.117 | // | // | // |
| PRE-H *vs.* POST-H | 0.475 | 0.930 | 0.488 | 0.802 | −0.644 | 0.03 |
| *Urine* | | | | | | |
| PRE-N *vs.* POST-N | 0.304 | 0.88 | −5.78·10⁻³ | // | // | // |
| PRE-H *vs.* POST-H | 0.254 | 0.932 | 0.754 | 0.72 | −0.472 | 4.58·10⁻⁶ |

[a]The models were considered valid only if the permutation test and *p-value* obtained from the cross validation ANOVA (CV-ANOVA) test ($p < 0.05$) were satisfied at the same time.

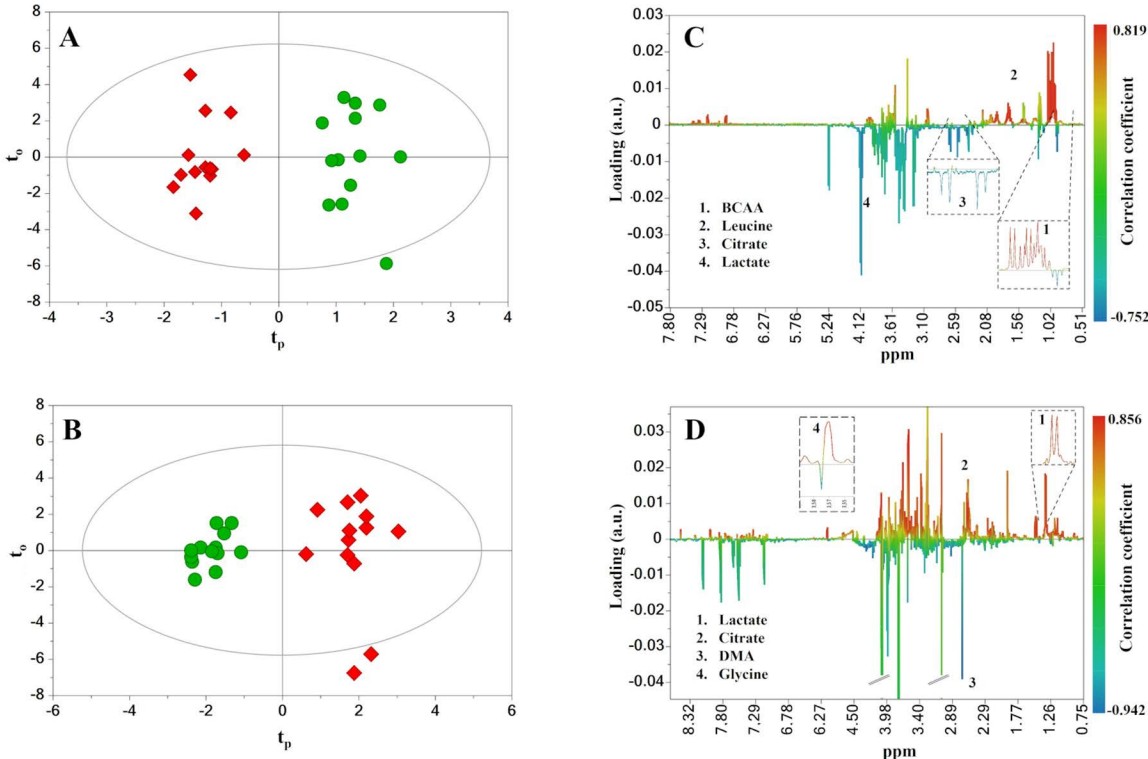

**Fig 3. OPLS-DA analysis of plasma and urine NMR datasets of HYPO session. (A)** Scores plot of plasma model. **(B)** Score plot of urine model. Symbols: ●, before session; ◆, after session. **(C)** Color-coded S-line plot of plasma model. **(D)** Color-coded S-line plot of urine model. The most important metabolites for the differentiation are labeled. Cutoff values for the identification of discriminant metabolites were p(cov) ≥ |0.05| and p(corr) ≥ |0.5|.

The model built with the plasma dataset explained approximately 47% of the variance in the X data matrix and 93% of the variance in the Y data set. The cross-validated $Q^2$ value was 0.488, indicating good predictive ability ($Q^2 > 0.4$). Additionally, the statistical robustness of the model was enforced by both permutation test and CV-ANOVA ($p < 0.05$). Comparable results were obtained from the OPLS-DA analysis of urine dataset. Indeed, the model exhibited acceptable values of explained variation related to classes ($R^2Y = 0.932$) and predictive ability ($Q^2 = 0.754$). Additionally, the CV-ANOVA test revealed that the model was reliable ($p < 0.0001$), and a permutation test for the Y variable indicated that it was not influenced by overfitting. The variables that had the most significant influence on the above-mentioned class separations were selected through the analysis of the correlation coefficient plots (Fig 3). The S-line plot of the plasma OPLS-DA model revealed five discriminant metabolites (Fig 3C): lactate, citrate, and branched-chain amino acids (BCAA: valine, leucine, and isoleucine). Regarding urine, the metabolites that most contributed to differentiate the pre and post-HYPO conditions were citrate, dimethylamine (DMA), glycine, and lactate (Fig 3D).

The impact of the oxygen exposure condition, exercise, and their interaction on plasma and urine metabolome were also investigated at individual metabolite level through two-way ANOVA. The overall results are presented in S1 and S2 Tables, for plasma and urine respectively. The findings suggested that exercise had a more pronounced effect on plasma metabolites compared to oxygen exposure alone, particularly on amino acids and key intermediates of energy metabolism. Among the most significant changes, isoleucine showed the largest effect from exercise ($p = 0.001$, $\eta^2 = 0.765$), indicating its significant role in the response to the performed physical activity. Other metabolites that showed significant modifications due to exercise included citrate ($p = 0.024$, $\eta^2 = 0.326$), leucine ($p < 0.001$, $\eta^2 = 0.592$), succinate ($p = 0.003$,

η²=0.542), tyrosine (p=0.01, η²=0.436), and valine (p=0.008, η²=0.455), all with large effect sizes. For other metabolites, no significant effects of exercise were found (p>0.05). Regarding oxygen exposure, only 3-hydroxybutyrate (3-HB) (p=0.019, η²=0.378) and carnitine (p=0.017, η²=0.392) exhibited significant changes. The interaction between exercise and oxygen exposure further highlighted the synergistic impact on several metabolite, including 3-HB (p=0.001, η²=0.602), isoleucine (p=0.014, η²=0.407), lactate (p<0.001, η²=0.677), leucine (p=0.027, η²=0.346), succinate (p=0.002, η²=0.570), and valine (p=0.023, η²=0.362). The significant differences (p<0.05) detected by the Bonferroni post hoc analysis are depicted as boxplots in Fig 4. As can be noted, these changes included an increase in the levels of citrate, lactate, succinate, and 3-HB in the samples collected after the HYPO test compared to PRE test samples. Conversely, the circulating levels of BCAA, phenylalanine, and tyrosine were more abundant in PRE samples. Additionally, specimens collected after the HYPO test displayed significantly greater levels of lactate, citrate, succinate, 3-HB, and carnitine compared to those taken after the NORMO test.

Concerning urine analysis, similar to the observations in plasma, the results indicated that physical exercise had a significant impact on various urinary metabolites, the oxygen exposure conditions further modulating these metabolic responses (S2 Table). The metabolites significantly affected by exercise were primarily related to energy metabolism and amino acid pathways, such as 3-hydroxyisobutyrate (3-HIB) (p=0.004, η²=0.511), 3-hydroxyisovalerate (3-HIV) (p=0.006, η²=0.484), acetone (p=0.008, η²=0.460), alanine (p=0.004, η²=0.506), glycine (p<0.001, η²=0.678), lactate (p<0.001, η²=0.630), and succinate (p=0.01, η²=0.440). A significant effect was also noted for dimethylamine (DMA) (p<0.001, η²=0.778) and taurine (p=0.005, η²=0.494). Regarding oxygen exposure, noteworthy changes were still observed in certain amino acids and energy-related metabolites, indicating distinct metabolic adaptations to hypoxic conditions. These alterations included 3-HIB (p=0.014, η²=0.405), alanine (p=0.016, η²=0.395), acetone ((p<0.001, η²=0.630), glycine (p=0.02, η²=0.578), lactate (p=0.05, η²=0.500), succinate (p<0.001, η²=0.629), and taurine (p=0.016, η²=0.396). The interaction between exercise and oxygen exposure conditions revealed a combined influence of these factors in a limited number of cases among with the most significant observed for acetone (p<0.01, η²=0.622), dimethylamine (p<0.001, η²=0.737), glycine (p<0.001, η²=0.611), and lactate (p=0.004, η²=0.514). Fig 5 shows the boxplots for the metabolites that exhibited significant differences following the Bonferroni correction. As can be seen, the contents of 3-HIB, 2-HIV, alanine, acetone, glycine, lactate, succinate and taurine increased following the HYPO test, while DMA levels decreased. Furthermore, acetone, DMA, glycine, lactate, and succinate showed a substantial difference in contents between samples collected after the NORMO and HYPO session.

Further insights on the changes occurring after the HYPO test were achieved by probing the inter-correlation between the aforementioned urine and plasma metabolites whose levels were significantly altered. Pearson pairwise correlation analysis was performed both separately for plasma and urine datasets and across the two biofluids. We considered correlation coefficients r≥|0.4| with a p-value<0.05 as statistically significant. Significant correlations were identified among plasma metabolites within the same compound classes (S1A Fig). Robust correlation clusters were observed among amino acids with a maximum correlation coefficient reaching 0.95 between isoleucine and leucine, indicating a very strong association among these metabolites. Additional noteworthy correlations included positive associations of citrate with both succinate (r=0.62) and lactate (r=0.73). Conversely, significant negative correlations were observed between organic acids and amino acids, ranging from r= −0.46 between citrate and leucine to r= −0.73 for the relation between citrate and tyrosine. Several correlations among urine metabolites were also identified (S1B Fig), with the strongest being between glycine and alanine (r=0.85) Both of these metabolites were positively correlated with 3-HIB (r=0.58 and r=0.56, respectively), 3-HIV (r=0.67 and r=0.63, respectively), lactate (r=0.73 and r=0.74, respectively), and acetone (r=0.59 and r=0.72, respectively). Lactate also showed positive correlations with 3-HIB (r=0.52) and 3-HIV (r=0.61). Conversely, urine DMA negatively correlated with glycine (r=−0.62), alanine (r=− 0.57), succinate (r=−0.53), and acetone (r=− 0.66). Fig 6 illustrates the heatmap for the correlations between the two biofluids.

As can be seen, plasma BCAA, phenylalanine, and tyrosine exhibited a positive correlation with urine DMA, and a negative correlation with urine succinate. Plasma BCAA negatively correlated also with urine lactate, 3-HIB, and 3-HIV.

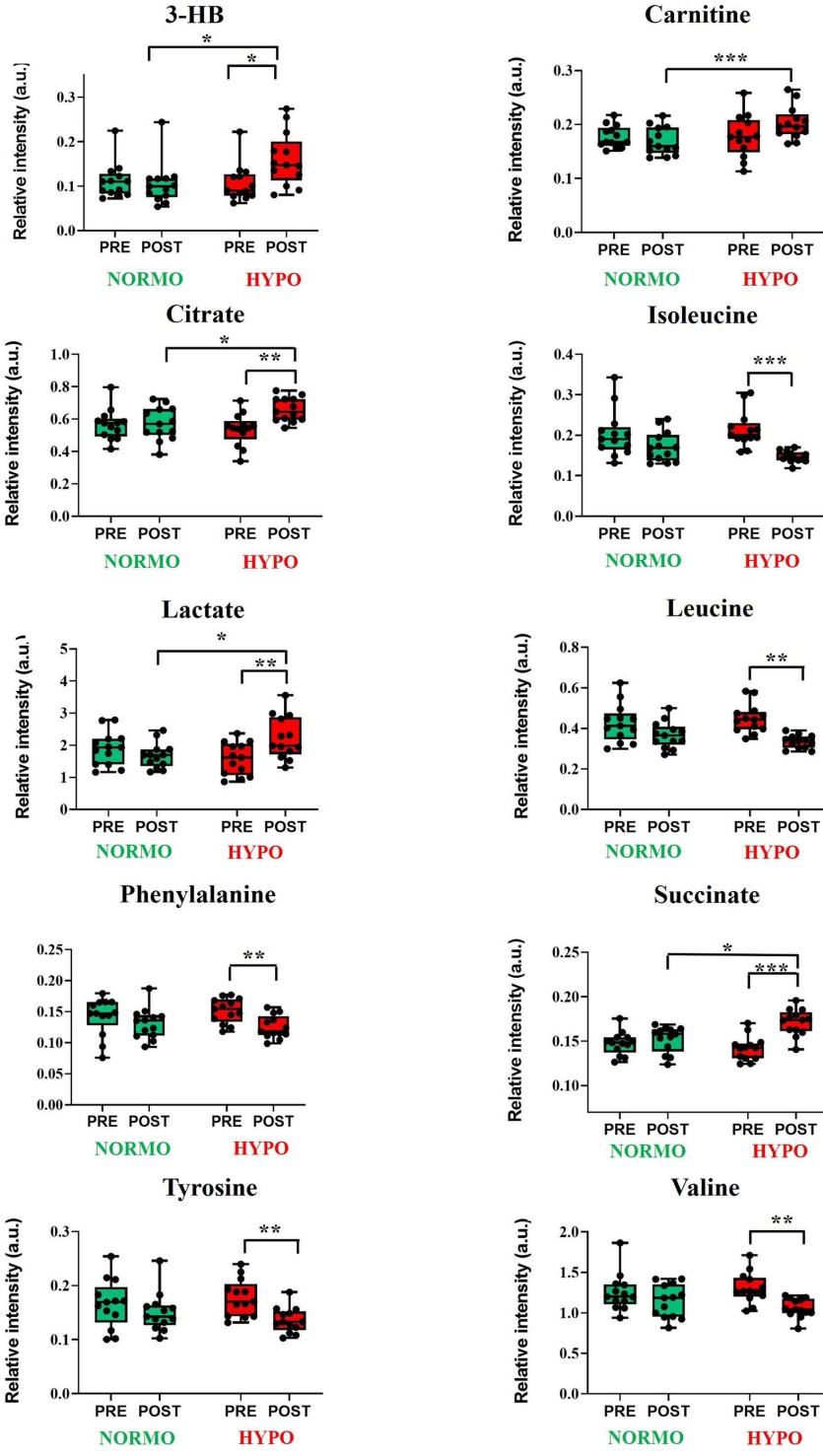

**Fig 4. Univariate statistical analysis of plasma metabolites.** Box plots representation of the significant differences in the levels of plasma metabolites before (PRE) and after (POST) the NORMO and HYPO tests. P-values < 0.05 were regarded as statistically significant: * p < 0.05; ** p < 0.01; *** p < 0.001.

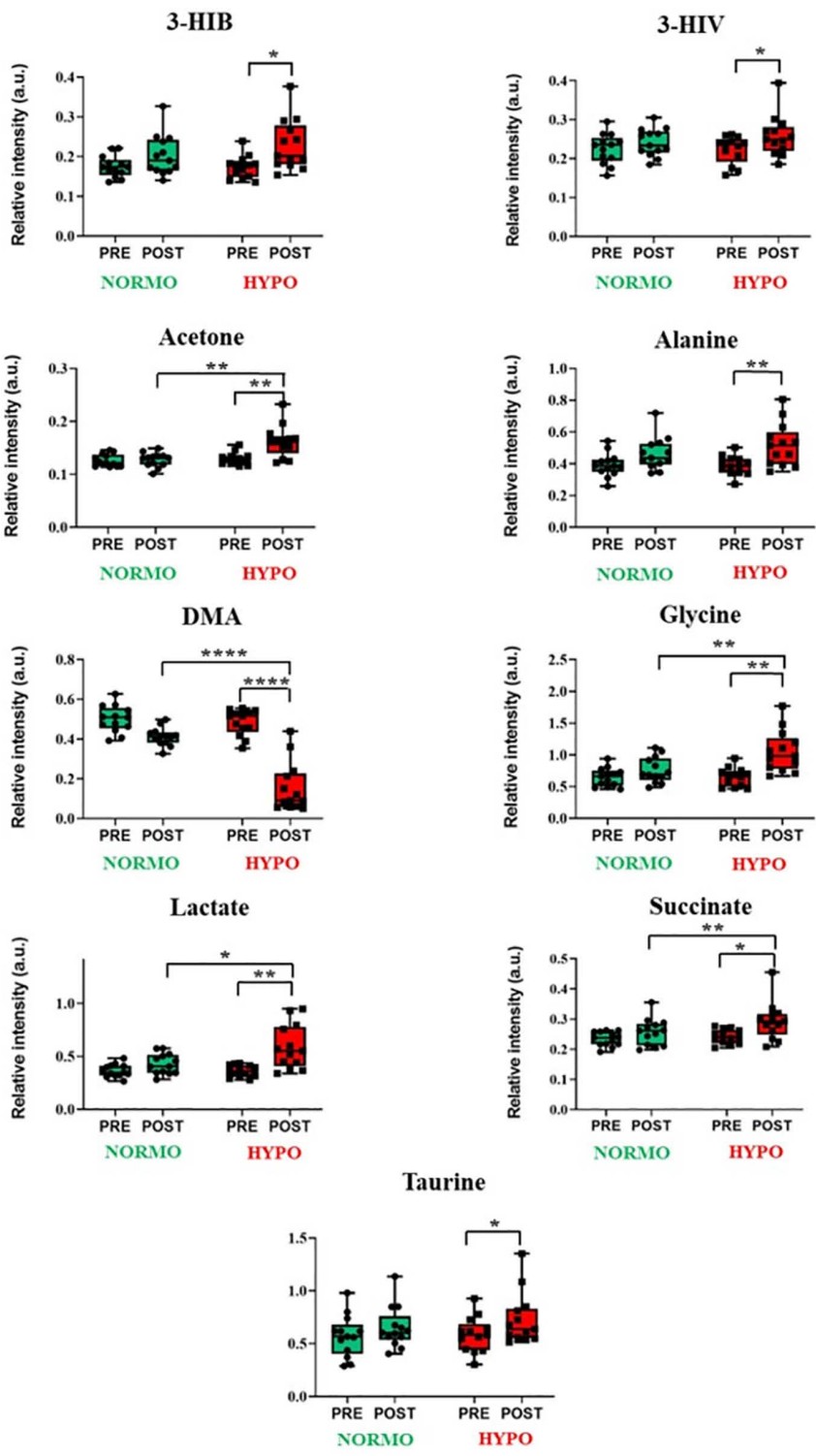

**Fig 5. Univariate statistical analysis of urine metabolites.** Box plots representation of the significant differences in the levels of urine metabolites before (PRE) and after (POST) the NORMO and HYPO tests. P-values < 0.05 were regarded as statistically significant: * p < 0.05; ** p < 0.01; *** p < 0.001; **** p < 0.0001.

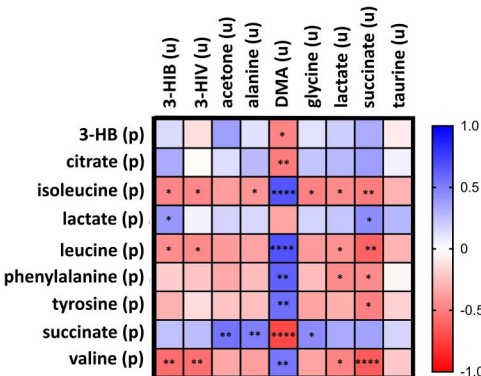

**Fig 6. Correlation analysis.** Heatmap illustrating the matrix of Pearson correlation coefficients of significantly altered plasma and urine metabolites after the HYPO test: blu, positive correlations; red, negative correlations. Metabolites are labeled with "(p)" for plasma and "(u)" for urine, respectively. Only statistically significant correlations with $|r| \geq 0.40$ and $p < 0.05$ are displayed. Significance levels: * $p < 0.05$; ** $p < 0.01$; *** $p < 0.001$; **** $p < 0.0001$.

Plasma succinate and citrate showed a negative correlation with DMA, while plasma succinate positively correlated with alanine, glycine, and acetone. Other positive correlations included plasma lactate with urine 3-HIB and succinate.

To further investigate the biological relevance of metabolite alterations, we examined whether changes in circulating and urinary metabolites were associated with the acute cardiovascular responses elicited by the exercise sessions. As shown in Table 2, most hemodynamic parameters exhibited significant and transient changes during the third minute of exercise, followed by a rapid return to baseline values during recovery. We therefore reasoned that this time point would be the most informative for exploring physiological–metabolic associations. Then, pairwise Spearman correlation coefficients were calculated between the metabolites measured pre- and post-exercise, and the hemodynamic parameters assessed during rest (pre) and at the third minute of exertion (Exe3). Only metabolites that showed statistically significant changes according to the above-mentioned ANOVA results were included in the correlation analysis, in order to focus on the most relevant features that responded to the experimental conditions. This strategy enabled us to investigate whether intra-subject shifts in metabolism align with the acute physiological stimulus triggered by exercise under different oxygenation conditions. Correlation analyses were performed separately for plasma and urine, and for NORMO and HYPO conditions. The results, summarized in the heatmaps of Fig. 7, revealed several significant correlations in the HYPO condition. For instance, in plasma (Fig. 7A) amino acids showed negative correlations with HR, CO, and SV, while correlating positively with SVR. Organic acids, including succinate and lactate, also showed meaningful associations. For instance, succinate and citrate levels positively correlated with HR, MAP and CO and negatively with SVR. Lactate was similarly correlated positively with HR and CO, and negatively with SVR. In urine (Fig. 7B), 3-HIB, 3-HIV, lactate, glycine, and acetone showed consistent positive correlations with HR and CO, and negative correlations with SVR. Additionally, DMA was inversely correlated with all cardiovascular parameters. Conversely, no statistically significant associations were detected under normoxic conditions, in either plasma or urine, indicating that the physiological stimulus of the NORMO test was not sufficient to generate detectable metabolic-hemodynamic correlations.

## Discussion

The present investigation examined the NMR-based metabolic profile of plasma and urine in thirteen physically active male participants who underwent two sessions of short-duration moderate dynamic exercise with varying minimum percentages of inspired oxygen. The NORMO test was conducted in normoxia ($FiO_2 = 21\%$), while the HYPO session was carried out in mild acute normobaric hypoxia ($FiO_2 = 13.5\%$), corresponding to an altitude of around 3500 m. Consistent with our earlier research [18], the hemodynamic responses of subjects to the NORMO and HYPO tests evidenced

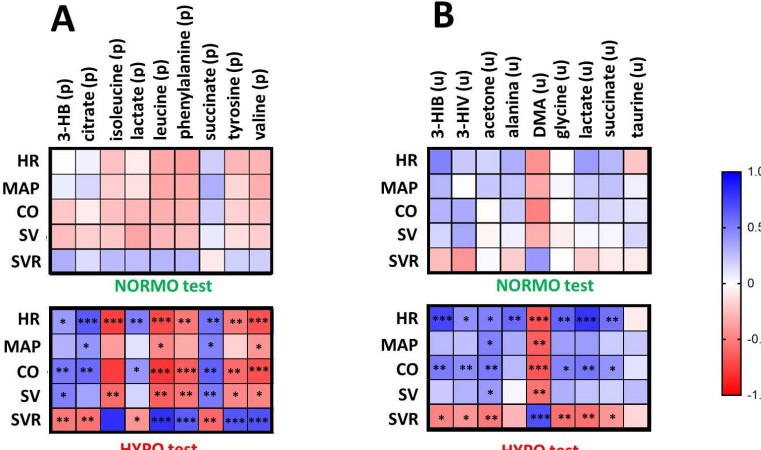

**Fig 7. Correlation analysis.** Heatmaps showing the matrix of Sperman correlation coefficients between hemodynamic parameters and metabolites in plasma (A) and urine **(B)**: blu, positive correlations; red, negative correlations. Metabolites are labeled with "(p)" for plasma and "(u)" for urine. Only statistically significant correlations with |r| ≥ 0.40 and p < 0.05 are displayed. Significance levels: *p < 0.05; **p < 0.01; ***p < 0.001. Abbreviations: HR, Heart rate; CO, cardiac output; MAP, mean arterial pressure; SV, stroke volume; SVR, systemic vascular resistance.

no discernible differences, indicating a similar tolerance of the circulatory system under both conditions. No significant changes were also noted in the subjects' urine and plasma metabolome following the NORMO session. It is well known that the metabolic and molecular responses to physical exercise are influenced by several factors. Among these, the intensity and duration of the exercise play crucial roles in determining the relative contribution of energy-generating pathways [28]. The short duration and low intensity of the exercise performed under our experimental conditions might provide a reasonable explanation for the lack of significant metabolic variations after the test performed in normoxia. Differently, significant modifications in the plasma and urine metabolome occurred under normobaric hypoxia, suggesting a molecular response to this specific condition. One of the most significant changes was an increase in lactate levels in both biofluids. Lactate is an intermediary for glycolytic metabolism and is closely tied to exercise intensity [29–33]. During moderate-intensity exercise, blood lactate concentration shows minimal increase compared to resting levels due to a balance between lactate oxidation and gluconeogenesis. However, as exercise intensity rises and oxygen supply becomes insufficient, anaerobic metabolism comes into play, leading to increased lactic acid production. The liver converts excess lactic acid into glucose through the Cori cycle, which is then metabolized back to lactate in skeletal muscles. This makes plasma lactate concentration a common indicator in exercise physiology to assess exercise or training intensity. It is also well-established that in an hypoxic environment, even in the absence of exercise, the body compensates for the reduction of oxygen availability by increasing anaerobic glycolysis and producing lactate as a byproduct. [34]. In the present study, the increase in lactate observed in both plasma and urine levels after the HYPO session suggests a metabolic response of non-competitive subjects to a mild dynamic exercise under conditions of reduced oxygen exposure. Therefore, even though the exercise was of low intensity, the reduced oxygen supply may have contributed to lactate accumulation, a phenomenon that could be accentuated in non-athletic individuals or those who are not competitively trained, as they may not have the same metabolic efficiency as trained athletes to handle hypoxic stress.

Other metabolites that exhibited significant changes following the HYPO test included citrate and succinate. Specifically, the levels of both metabolites increased in plasma, whereas in urine, only succinate content rose. Notably, although we did not detect changes in both metabolite contents after the NORMO test, their levels at the end of this session were significantly lower compared to those measured after the HYPO test. Citrate and succinate are intermediates of the tricarboxylic acid (TCA) cycle, a crucial energy-producing system in mitochondria. This cycle is typically activated during

intense continuous exercise in response to an increased energy expenditure associated with rising exercise volume and intensity. Consequently, the observed accumulation of TCA intermediates in the bloodstream is taken as an indication of heightened activity in this metabolic pathway [35]. Succinate, in particular, has also been identified as a metabolic signal that becomes especially relevant under hypoxic stress [36]. Under such circumstances, it may accumulate in tissues and blood, potentially contributing to cellular responses through changes in mitochondrial function. Interestingly, a recent metabolomics study has reported changes in plasma succinate during a concentric-eccentric leg exercise performed at a simulated altitude of 3.500 m, attributing these changes to hypoxic stress [12]. Moreover, succinate has been suggested to influence vascular function, possibly promoting vasodilation to facilitate oxygen delivery during metabolic stress [37]. Consistent with this hypothesis, our results revealed significant correlations between succinate levels (in both urine and plasma) and hemodynamic parameters. In particular, the observed inverse correlation with SVR strengthens the idea of a possible active role of succinate in cardiovascular adaptation mechanisms under hypoxic conditions. Although the functional implications of changes in plasma and urine TCA intermediates after the HYPO test cannot be conclusively determined at this stage based solely on our findings, the significant variations in succinate and citrate levels following the hypoxic session, alongside their correlation with cardiovascular responses, points to their potential involvement in the integrated systemic adaptation to hypoxic exercise.

It is well known that in conditions of a relative lack of energy, such as intense exercise or hypoxia, changes in free amino acid (AA) content can take place in skeletal muscle to meet physiological demands [38–39]. Glucogenic AA are used for gluconeogenesis, while ketogenic AA can generate ketone bodies via acetyl coenzyme A or are simply metabolized via the TCA cycle. Branched-chain amino acids (BCAA), i.e., valine, leucine and isoleucine, contribute little to energy supply unless carbohydrate availability is low. Leucine is a ketogenic amino acid used by the liver to synthesize ketone bodies, while isoleucine and valine are both ketogenic and glucogenic. In response to physical activity [40–42] or hypoxic stress [10], BCAA utilization is known to increase, supporting energy production or other biosynthetic pathway and facilitating adaptation to low-oxygen conditions. In the current investigations, we observed a significant decrease in the plasma BCAA levels of participants after the HYPO test, with a discriminant role between PRE-H and POST-H groups. Differently, no significant changes were noted after the NORMO test, highlighting a specific metabolic adaptation to the hypoxic challenge. This interpretation is further supported by our correlation analysis which revealed significant negative associations between plasma BCAA levels and hemodynamic parameters such as cardiac output and heart rate, along with positive correlations with systemic vascular resistance. These patterns suggest that greater cardiovascular strain during hypoxic exercise may be accompanied by enhanced BCAA catabolism. Moreover, we found a negative correlation of BCAA with 3-hydroxyisobutyrate (3-HIB) and 3-hydroxyisovalerate (3-HIV), both metabolic intermediates of BCAA degradation. Collectively, these findings support the hypothesis that BCAA metabolism contributes to meeting energy demands and cardiovascular adaptation in non-athletic subjects under our hypoxic experimental conditions.

Phenylalanine and tyrosine in plasma followed a similar pattern as BCAA, experiencing a decrease after the HYPO test. These aromatic AA possess both glucogenic and ketogenic properties. Additionally, phenylalanine acts as a precursor for the synthesis of catecholamines [43]. Catecholamines are well known for the role played in body's physiological adjustment to a variety of physical, environmental, and behavioral stressors, including environmental hypoxia and exercise [44–46]. Although the current study did not measure plasma catecholamine concentrations, we do not exclude the possibility that the reduction in phenylalanine and tyrosine levels under the hypoxic conditions of the HYPO test may be linked to elevated levels of circulating adrenaline. Additionally, it is noteworthy that a recent metabolomics investigation has highlighted phenylalanine metabolism as one of the most prominently altered pathways under high-altitude exposure [47]. Thus, a potential alternative explanation for our observations remains plausible.

Alterations after the HYPO session also included an increase in urine glycine and alanine levels. These changes were positively correlated with one other. Glycine is an amino acid capable of serving as a substrate in gluconeogenesis. Experimental evidence has highlighted significant changes in this metabolite following physical activity. However, the

findings haven't been always consistent. Indeed, higher urinary excretions of glycine have been observed 30 minutes after acute physical exercise [32] as well 2 h after high-intensity interval and continuous moderate exercise [48]. In contrast, a decrease in glycine levels has been noted after high-intensity exercises in athletes aged 50–60 years [49]. To the best of our knowledge, no previous evidence of glycine changes after a mild dynamic exercise under hypoxia has been reported. Alanine plays a pivotal role in glucose metabolism through the glucose-alanine cycle, favoring the use of amino acids by the skeletal muscle as an additional source of energy, while ensuring a continuous cycling of nutrients from the muscle to liver. This mechanism contributes significantly to energy homeostasis. In our HYPO test, alanine correlated positively with plasma succinate and negatively with plasma isoleucine. These associations may imply that during this exercise section, metabolic processes in the liver integrated pathways of carbohydrate and amino acid metabolism to provide substrates for muscular energy metabolism.

Another metabolite that showed significant alterations after the HYPO test was taurine. Taurine, an essential amino acid, plays a crucial role in maintaining cellular osmotic balance, protecting against oxidative stress, supporting mitochondrial function, and regulating calcium ion handling during muscle contraction. [50]. Numerous studies have shown that plasma taurine levels increase during exercise, but the precise mechanism underlying this rise still remains under investigation. [51,52]. While we cannot ascertain the precise nature of the increase in urinary taurine levels after the HYPO test, we acknowledge the possibility that this change may reflect a biological response of non-athletic individuals to exercise under normobaric hypoxia. However, further investigations are necessary to fully understand the underlying mechanisms of this response and its potential implications for metabolic adaptations to hypoxia.

Similarly to taurine, the role of the lower content of urine dimethylamine (DMA) in POST-H compared to PRE-H remains uncertain. DMA showed a negative correlation with plasma citrate and succinate and a positive correlation with plasma BCAA, phenylalanine, and tyrosine. As a metabolite produced through bacterial metabolism, DMA levels, typically, reflect variations in the metabolic status of human intestinal bacteria. Recent findings suggest that changes in DMA may be linked to muscular stress [53]. Since the reasons behind the alterations in DMA levels following the HYPO test are not understood at this stage, additional studies may be required for further comprehensive investigation into potential associations of this metabolite with physical activity and/or hypoxia.

Lastly, we wish to highlight two other interesting findings: an increased content of plasma 3-hydroxybutyrate (3-HB) and urine acetone in POST-H compared to PRE-H. Both metabolites are ketone bodies produced from fatty acid oxidation. Regarding 3-HB, a slight rise in its levels has been shown to have physiological effects under certain circumstances, potentially influencing signaling pathways to address cellular stress in tissues and organs, including hypoxia-induced stress [54]. Thus, it is likely that our experimental observations may stem from a signature of fatty acid oxidation in individuals stimulated by hypoxia. This hypothesis could also account for the higher levels of carnitine in POST-H relative to POST-N, as carnitine plays a crucial role as a co-factor in facilitating the transport of fatty acids to the mitochondrial matrix for breakdown and utilization in energy production. Interestingly, correlation analysis revealed that 3-HB levels in plasma were positively associated with heart rate and cardiac output and inversely correlated with systemic vascular resistance. Similarly, urinary acetone showed a significant negative correlation with SVR. Growing evidence supports the notion that ketone bodies serve not only as alternative energy substrates but also as signaling molecules involved in cardiovascular regulation [55]. Taken together, these observations suggest that ketone metabolism may contribute actively to the systemic physiological adjustments elicited by hypoxic stress, beyond simply reflecting altered substrate utilization.

## Study strengths and limitations

Our study has several limitations. The small sample size is a primary drawback, potentially compromising the validity and reliability of results and constraining the exploration of intricate interactions or relationships between variables. However, a post-hoc power analysis, based on the η² values derived from the ANOVA tests for the most significantly altered metabolites, indicated that the study had adequate power (>80%) to detect the effects of exercise and its interaction with

oxygen availability on plasma metabolites, supporting the interpretation that hypoxia contributes meaningfully to metabolic changes when combined with physical activity. For the main effect of oxygen exposure alone, the power was slightly lower (75%), suggesting a moderate risk of Type II error (S2 Fig), and highlighting the value of future studies with larger samples to confirm these trends. Differently, the post hoc analysis for urinary data demonstrated excellent statistical power (>90%) for all three tested effects (exercise, oxygen exposure, and their interaction), ensuring the robustness of these findings. It is worth mentioning that even studies with limited sample sizes can yield insightful preliminary results, serving as a foundation for testing hypotheses that can be subsequently validated by larger studies or meta-analyses. Thus, our findings have the potential to contribute to advancing research and identifying crucial variables or patterns relevant to studies on the metabolic response of non-competitive subjects to hypoxia.

The homogeneity of our study population poses another limitation. Despite efforts to control for various factors such as age, BMI, and gender, the uniformity of participant characteristics may restrict the generalizability of our results to a more diverse and extensive population.

It might be also argued that the relative intensity of exercise under hypoxia was higher compared to that in normoxia, since both procedures employed the CPT under normoxic settings as a reference point to assess exercise load. Consequently, this makes more challenging for us to discern whether the metabolic alterations observed after the HYPO test were attributable to a higher relative effort, the hypoxic environment, or a combination of both. Nonetheless, it is important to highlight that all subjects exhibited similar heart rates under both normoxic and hypoxic conditions. This finding suggested similar cardiovascular activation and lead us to dismiss differences in the level of effort between the two workout scenarios.

## Conclusions

To the best of our knowledge, this study is the first to employ metabolomics to analyze two biofluids, namely urine and blood, in order to assess the effects of combined hypoxia and exercise in non-athletic individuals. Our preliminary results demonstrate that metabolism of these subjects is highly responsive to an acute bout of mild dynamic exercise performed under normobaric hypoxia. Indeed, the NMR analysis conducted on blood and urine metabolome revealed significant differences in the level of several metabolites between normoxic and hypoxic conditions, showing that participants exhibited notable metabolic responses to hypoxia even at relatively low workloads. These significant modifications point to shifts in bioenergetic metabolic pathways and a possible activation of a biological response to hypoxic stress. Key metabolites, including amino acids (alanine, glycine, branched-chain amino acids, tyrosine, and phenylalanine), lactic acid, intermediates of the Krebs cycle (citrate and succinate), metabolites of amino acid degradation (3-hydroxyisobutyrate and 3-hydroxyisovalerate), and osmolytes (taurine) reflect these metabolic adaptations.

Notably, although overall hemodynamic responses did not significantly differ between normoxic and hypoxic tests, we observed strong and coherent correlations between metabolite levels and cardiovascular variables only under hypoxia. This suggests that metabolomic profiling may capture physiological stress responses that remain undetected through standard cardiovascular measures. Even modest cardiovascular alterations appeared sufficient to trigger detectable changes in metabolic pathways related to energy balance, substrate preference, and redox regulation. Furthermore, several of the metabolites altered in response to the hypoxic challenge are also known to be dysregulated in cardiometabolic diseases [37,56,57]. While our participants were healthy, this overlap suggests that the observed metabolic shifts may reflect general homeostatic mechanisms that become impaired in disease, highlighting the relevance of our findings to both physiological and pathophysiological contexts.

It is worth noting that the effectiveness of the adaptation of non-competitive individuals to hypoxia may not be on par with that of competitive athletes, who likely benefit from greater physiological plasticity due to specific training regimens. Nevertheless, the consistent metabolic changes observed in our moderately active participants suggest that even low-intensity hypoxic exercise can engage relevant regulatory pathways. Together, these observations highlight the potential

of metabolomics as a powerful tool for designing personalized hypoxia-based protocols aimed not only at improving performance, but also at promoting recovery and metabolic health in clinical and rehabilitative settings.

## Supporting information

**S1 Fig. Correlation analysis.** Heatmap illustrating the matrix of Pearson correlation coefficients of significantly altered plasma (A) and urine (B) metabolites after the HYPO test: blu, positive correlations; red, negative correlations. * $p < 0.05$; ** $p < 0.01$; *** $p < 0.001$; **** $p < 0.0001$.
(PDF)

**S2 Fig. Post-hoc power analysis related to repeated measures ANOVA on plasma metabolites.** The figure illustrates the estimated statistical power as a function of total sample size, based on a post-hoc analysis conducted using G*Power. The analysis refers to the main effects of exercise, oxygen exposure, and their interaction. Effect sizes ($\eta^2$) were computed as the average of those obtained for the metabolites that showed statistically significant differences in the repeated measures ANOVA.
(PDF)

**S1 Table. Two-way ANOVA analysis of plasma metabolite levels.** Factors: oxygen exposure condition and exercise. Statistical significance was set at p-value $< 0.05$. Partial eta squared ($\eta^2$) values indicate effect sizes.
(PDF)

**S2 Table. Two-way ANOVA analysis of urine metabolite levels.** Factors: oxygen exposure condition and exercise. Statistical significance was set at p-value $< 0.05$. Partial eta squared ($\eta^2$) values indicate effect sizes.
(PDF)

## Acknowledgments

All authors wish to thank Prof. Antonio Crisafulli, the principal investigator of this project to whose memory this article is dedicated. Antonio was a distinguished and brilliant colleague, as well as a dear friend. His absence is deeply felt, and we miss him profoundly.

## Author contributions

**Conceptualization:** Flaminia Cesare Marincola, Raffaella Isola, Romina Vargiu, Elisabetta Marini, Silvana Roberto, Gabriele Mulliri, Antonio Crisafulli, Andrea Rinaldi.

**Data curation:** Giovanna Ghiani, Gabriele Mulliri.

**Formal analysis:** Flaminia Cesare Marincola, Daniela Masu, Veronica Libonati, Michela Tozzi.

**Funding acquisition:** Antonio Crisafulli, Andrea Rinaldi.

**Investigation:** Daniela Masu, Veronica Libonati, Sara Magnani.

**Methodology:** Flaminia Cesare Marincola, Gabriele Mulliri, Antonio Crisafulli.

**Resources:** Flaminia Cesare Marincola, Antonio Crisafulli.

**Supervision:** Flaminia Cesare Marincola, Gabriele Mulliri, Antonio Crisafulli.

**Validation:** Flaminia Cesare Marincola.

**Writing – original draft:** Flaminia Cesare Marincola.

**Writing – review & editing:** Flaminia Cesare Marincola, Daniela Masu, Veronica Libonati, Raffaella Isola, Romina Vargiu, Elisabetta Marini, Silvana Roberto, Gabriele Mulliri, Andrea Rinaldi.

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
