## [Decision Letter · Decision Letter 0]

Dear Dr. Cesare Marincola,

Thank you for submitting your manuscript to PLOS ONE. After careful consideration, we feel that it has merit but does not fully meet PLOS ONE’s publication criteria as it currently stands. Therefore, we invite you to submit a revised version of the manuscript that addresses the points raised during the review process.

**ACADEMIC EDITOR:**

We look forward to receiving your revised manuscript.

Kind regards,

Domingo Jesús Ramos-Campo, Ph.D

Academic Editor

PLOS ONE

Additional Editor Comments:

Please, review the article according to the reviewer's comments

Reviewers' comments:

Reviewer's Responses to Questions

**Comments to the Author**

1. Is the manuscript technically sound, and do the data support the conclusions?

Reviewer #1: Partly

Reviewer #2: Yes

2. Has the statistical analysis been performed appropriately and rigorously?

Reviewer #1: Yes

Reviewer #2: Yes

3. Have the authors made all data underlying the findings in their manuscript fully available?

Reviewer #1: No

Reviewer #2: Yes

4. Is the manuscript presented in an intelligible fashion and written in standard English?

Reviewer #1: Yes

Reviewer #2: Yes

Reviewer #1: This is an interesting metabolomic study on the effect of short-time (3-min) low-intensity exercise. The metabolic response (plasma and urine) was assessed using robust up-to-date methods, which is the study's strongest point. The measurement procedures seem to be planned and executed properly. I fully appreciate the work done and the meticulous analysis.

However, there is a serious problem with terminology, resulting in huge over- and misinterpretation of the obtained data, partly inadequate discussion, and exaggerated unjustified conclusions. The Authors, throughout the manuscript, continually confuse “(competitive) sport” with “leisure time physical/sporting activity”, “athletes” with “non-competitive leisure exercisers”, and, finally “training” with “exercise”. Below I try to explain key terms.

General comments

The definition for competitive athlete as it is suggested to be used in, but not limited to, health and medical science research is available here: https://www.acc.org/latest-in-cardiology/articles/2016/06/27/07/06/the-terms-athlete-and-exercisers or here: Scand J Med Sci Sports. 2016 Jan;26(1):4-7. doi: 10.1111/sms.12632. Briefly, there are several criteria, which must ALL be fulfilled to define someone as an athlete (performance improvement, participating in sport competition, formally registered in a sport federation, sport/competition as major activity and focus exceeding the time allocated to other professional or leisure activities). In this light, the study participants are not athletes, they probably meet or are above the minimal WHO criteria included in the guidelines on physical activity and sedentary behavior. They just participate in leisure time sporting/physical activity, which refers to all of the behavior connected with physical activity that people engage in their freely disposable time. That they are not competitive athletes may be also concluded from their maximum power achieved at VO2max (255 W; athletes typically exceed 400 W). Also, their VO2max is rather low or moderately above the population average at most (about 30 or 70 percentile of normal population, depending on assignment to age group; see e.g. BMJ Open 2018, 8:e018697, doi: 10.1136/bmjopen-2017-018697; Mayo Clin Proc 2022, 97(2):285-293, https://doi.org/10.1016/j.mayocp.2021.08.020). Since the research presented here does not involve athletes, the authors should not comment or speculate on it, and in particular cannot call their subjects athletes.

In addition, the exercise intensity used was 30% of Wmax attained during the preceding test until exhaustion. In terms of competitive sports (and not only), it is a very low intensity. This is much lower than lactate threshold in athletes (~64-84% of Wmax, depending on the assessment method; see e.g. BMC Sports Sci Med Rehabil 2020, 12, 70, https://doi.org/10.1186/s13102-020-00219-3) and probably even lower than the intensity at lactate threshold in healthy non-athletic adults (was any metabolic threshold determined in this study?). 30% of Wmax in normoxia means the lowest intensity zone used in endurance training, called ‘active recovery’ (or ‘very light’). I would venture the hypothesis that professional athletes, even in hypoxia, would remain in this lowest zone at an intensity of 30% Wmax and a significant metabolic response would hardly occur. Especially since the duration of this single exercise (3 minutes) at such a low intensity is too short to be an effective exercise stimulus for competitive athletes. As is cited I the introduction, other research teams used exercise under hypoxia lasting several tens of minutes (and of higher intensity).

Anyway, conclusions for competitive sports and training cannot and most not be drawn from the research presented. Even if one assumes that certain metabolic mechanisms are universal, the metabolic adaptations of the competitive athlete (plus inborn predispositions) usually yield a metabolic profile extremely different (quantitatively and qualitatively) from that of the ‘average’ population as numerous studies show. In addition, there are fluctuations in adaptive metabolic changes during successive phases of the annual training cycle (e.g., general preparation, special preparation, pre-competition/taper phase, competition phase). By the way, had I read only the last sentence of the main text (lines 552-3), I would have erroneously concluded that the subjects were elite athletes being followed in a long training cycle. This is misleading and completely impossible on the basis of this research!

Training is a broad term, however, it always refers to a longer systemic PROCESS of developing skills, knowledge or, as in athletes, mainly (but not only) physical fitness and performance. The concept of exercise, as it is used in this study in a narrow physiological sense, means only a single bout of effort accompanied by resulting metabolic response. It takes at least several weeks or even months of planned and systematically repeated sessions and series of specific exercise bouts constituting the (physical) training to yield adaptive changes. The presented study does not involve any long-term process and therefore the authors cannot and must not comment on it, in particular interpret the results in the context of sports training.

To sum up, the manuscript should therefore be thoroughly rewritten: unwarranted references/comments/speculations/conclusions related to competitive sports, athletes and training should be removed, and instead content related to the response to a single low-intensity exercise bout in healthy young adults should be presented. Below is the list of places were the unwarranted terms were used:

athlete(s): Lines 27, 49, 53, 62, 91, 113, 261, 276, 406, 413, 415, 482, 510, 529, 539, 543, 553

sport(s): Lines 58, 71, 73, 107, 259

training: Lines 26, 55, 58, 60, 73, 426, 468, 484, 520, 534, 540, 542, 545, 553

The most interesting observation to me from this study is that significant metabolic changes can be induced in a healthy young person under hypoxia even with surprisingly low-intensity and short-duration exercise (one that does not induce changes in normoxia). Please try to propose/discuss any application of these results in a healthy non-athletic population. Could it be an aid to recreational training? Replacement of any form/modality of exercise? Pre-adaptation for something? Health prevention? Rehabilitation? In what cases/situations? (instead of speculating on competitive athletes).

Other comments

Can you please explain/justify why an intensity of 30% Wmax and an exercise duration of 3 min was used?

Table 1. Please provide the following values: SV, HR, CO, MAP, SVR that were measured, as revealed in the ‘Hemodynamic assessment’ section. Or remove the section if not relevant to the results presented.

When, on which day, was the screening CPET performed? The same or other day as experimental trials?

Was lactate or ventilatory threshold determined during the screening CPET? Pleas provide parameters in Table 1 if available. Probably the participants did not reach the threshold in normoxia. Interesting, was the lactate/ventilatory threshold achieved or exceeded in hypoxia?

I think, the values of basic cardiorespiratory parameters and lactate for normoxia and hypoxia should be provided and compared (HR, VO2, VE, etc.) in a separate table for general characteristic. Could you also provide peak lactate values for normoxia and hypoxia?

Line 162‒3 This was recovery between the normoxia and hypoxia tests? Unclear.

After the experimental trials, what was collected first? Blood or urine? Could the time from trial termination to sampling affect the results?

Univariate analysis (2-way ANOVA). Please always provide effect size (eta-squared) in the main text and supplementary material, apart from p-value (that itself does not inform of the strength if the effect). Also, please first inform about the main effects (condition, pre-post and their interaction) before providing post hocs.

First of all, please revise the introduction and discussion, taking into account my earlier comments and suggestions. Please also make an adequate selection of literature cited. Please be more considerate and conservative in your interpretations, which must be based on the results obtained in a specific group of people (who are not athletes).

Reviewer #2: Congrats on a very well organized and methodically prepared study. Correctly selected statistical methods, sczegosely described research results. While you can not have comments on the methodological side, in the very purpose and interpretation of the study in my opinion the authors made a mistake. An error that affects the discussion, which is written correctly. Unfortunately, it does not fully correspond to the introduction and assumptions of the study. reading the work, you get the impression that the authors are introducing the problem of altitude hypoxia. Hypoxic stimulus is then associated with changes in oxygen partial pressure. In the paper we have an experiment based on normobaric hypoxia. In both cases of application of hypoxia, the hypoxic effect is different. While in the first case we can expect an increase in the aerobic component of the effort, in the second case it is not surprising to see an increase in the anaerobic component. Very important is also the susceptibility of the athlete's body to the application of a hypoxic stimulus. In sports training, one must be prepared for hypoxic exposure. The VO2max values of the subjects are so low that they are rather indicative of individuals with a serious deficit in aerobic capacity. The state of preparation in this respect of the study group should be more specifically considered in the discussion. In the reviewer's opinion, the paper needs to be supplemented/revised with an introduction to the mechanisms of normobaric hypoxia. It is also necessary to introduce into the discussion elements specific to the effects of using nrmobaric hypoxia.

**Do you want your identity to be public for this peer review?** For information about this choice, including consent withdrawal, please see our Privacy Policy

Reviewer #1: **Yes: ** Krzysztof Kusy

Reviewer #2: **Yes: ** Tomasz Gabrys

---

## [Author Response · Author response to Decision Letter 1]

8 Oct 2024

Dear Prof. Ramos-Campo,

We are pleased to submit a revised version of our manuscript titled Metabolic response to an acute bout of mild dynamic exercise performed under normobaric moderate hypoxia: a NMR-based metabolomics study. We have carefully addressed journal’s requirement (see page 2) and reviewer’s comments

We greatly appreciate the confidence you have shown in our study. We would like also to extend our sincere thanks for the valuable comments and suggestions provided by the reviewers. Their comprehensive reviews and insightful feedback have been instrumental in improving the quality and clarity of our work, guiding us through the enhancement of the manuscript.

Our responses to the reviewers’ comments and criticisms (in blue) are provided in a dedicated file.

Sincerely yours,

Flaminia Cesare Marincola

---

## [Decision Letter · Decision Letter 1]

Dear Dr. Cesare Marincola,

Thank you for submitting your manuscript to PLOS ONE. After careful consideration, we feel that it has merit but does not fully meet PLOS ONE’s publication criteria as it currently stands. Therefore, we invite you to submit a revised version of the manuscript that addresses the points raised during the review process.

We look forward to receiving your revised manuscript.

Kind regards,

Domingo Jesús Ramos-Campo, Ph.D

Academic Editor

PLOS ONE

Additional Editor Comments:

Please, review the article according to the reviewer's comments

Reviewers' comments:

Reviewer's Responses to Questions

**Comments to the Author**

Reviewer #1: Minor Revision

Reviewer #2: Major Revision

2. Is the manuscript technically sound, and do the data support the conclusions?

Reviewer #1: Yes

Reviewer #2: Yes

3. Has the statistical analysis been performed appropriately and rigorously?

Reviewer #: 1Yes

Reviewer #2: Yes

4. Have the authors made all data underlying the findings in their manuscript fully available?

Reviewer #1: No

Reviewer #2: Yes

5. Is the manuscript presented in an intelligible fashion and written in standard English?

Reviewer #1: Yes

Reviewer #2: Yes

Reviewer #1: Manuscript Number: PONE-D-24-21080R1

Title: Metabolic response to an acute bout of mild dynamic exercise performed under normobaric moderate hypoxia: a NMR-based metabolomics study

Authors claims this is the first study to use metabolomics on both urine and blood plasma to assess the combined effects of hypoxia and exercise in non-athletes. Thirteen healthy young adult participants performed identical sessions of mild exercise (3 minutes at 30% Wmax) under both normal oxygen (normoxia) and normobaric hypoxia (simulated altitude) conditions. Blood and urine samples were collected before and after exercise, and analyzed using 500 MHz NMR spectroscopy. Results showed significant metabolic response to mild exercise under hypoxia, even at low workloads, highlighting the potential of hypoxic exercise for non-athletes. Plasma analysis revealed decreased branched-chain amino acids, phenylalanine, and tyrosine, alongside increased lactate, succinate, citrate, and 3-hydroxybutyrate. Urine analysis showed increased 3-hydroxyisobutyrate, 3-hydroxyisovalerate, alanine, lactate, succinate, acetone, taurine, and glycine, with decreased dimethylamine. These changes suggest alterations in energy metabolism pathways and potential metabolic stress.

Overall, the study emphasizes the potential of metabolomics for understanding metabolic adaptations of non-athletic individuals to hypoxic conditions. This is crucial for developing effective rehabilitation and health strategies to improve recovery and optimize health outcomes in non-athletic populations. The revised manuscript is well written and I recommend the publication of the article with a few minor suggestions and corrections to further enhance the manuscript's quality

Specific comments:

Comment: Authors used CHENOMX for assignment purpose, however, it can also be used for concentration profiling. Specifically, the circulatory levels of discriminatory metabolic features can be profiled using CHNEOMX considering format as an internal reference for blood plasma samples and creatinine for urine samples. The results corresponding to statistical comparison can be discussed in the main manuscript and Figures can be included in the supplementary materials for future reference.

Comment: The statistical correlation analysis between marker metabolic features with key clinical parameters need to be performed, if possible. The statistically significant correlation will establish the biological relevance of this study. The correlation analysis needs to performed separately for normoxia and hypoxia groups, accordingly there will be two correlation heat maps for plasma samples and two correlation heat maps for urine samples. The significance of the statistical correlation is expected to be high for discriminatory metabolic features in Hypoxia group. Authors may refer to this paper for correlation analysis:

BioCPR–A Tool for Correlation Plots (https://www.mdpi.com/2306-5729/6/9/97)

Comment: For future validation studies and establish the reproducibility of the method, authors should submit all the NMR raw data to a public repository like metabolights of workbench or ZENODO database. The corresponding details should be incorporated into the manuscript as a data availability statement.

Reviewer #2: Dear Authors,

I had the opportunity to review the revised version of your manuscript, which investigates the effect of a brief exercise bout in normobaric hypoxia on metabolic and physiological variables in physically active young adults. The relevance of this topic is notable, given the growing interest in hypoxia strategies to optimize health and performance. However, I would like to respectfully suggest some areas for improvement. Specifically, the description of the study design could be clarified, and it would be beneficial to define the type of investigation (whether experimental and/or a clinical trial). Additionally, the organization of the Materials and Methods section could be streamlined.

To address these points, a comprehensive evaluation of the submitted manuscript is warranted. This evaluation encompasses methodological robustness, data presentation, statistical analyses, and clarity of reporting:

1. The study design, which centers on the impact of brief exercise under moderate normobaric hypoxia, is conceptually relevant. The collection of metabolic and physiological measures, along with the statistical comparisons demonstrating consistent differences between normoxic and hypoxic conditions, is noteworthy. However, your manuscript would benefit from a clearer classification of its design, such as whether it is best described as an experimental crossover trial or another format. While the data presented support the general conclusions, the findings would be more robust if you explicitly detailed the randomization and enrollment procedures, the rationale for sample size, and the assumptions underlying the statistical tests performed.

2. Additionally, strengthening the link between specific metabolite changes and their physiological significance would enhance the discussion. The consistency between plasma and urine findings is intriguing; however, your manuscript should establish how these results connect with broader physiopathological or performance-related frameworks.

3. Your statistical approach is generally appropriate for investigating acute exercise effects. However, the manuscript would benefit from greater clarity regarding normality tests (e.g., Shapiro–Wilk), corrections for multiple comparisons (if multiple metabolites were tested simultaneously), and effect size metrics. While your results indicate significant changes, you should confirm that the sample size is sufficient to detect the observed effect sizes with adequate power. Additionally, explicitly stating how missing data or potential outliers were handled would reinforce the transparency of your study.

4. From a linguistic and structural standpoint, your manuscript is generally well-written and understandable. The English is clear enough for an international audience, though certain sections could be condensed or reorganized for improved readability. Specifically, the Materials and Methods section would benefit from a more conventional structure, beginning with the study design and participant flow, followed by details on the exercise protocol, biological sampling, and analytical methods. Improving the resolution of Figure 1 and ensuring that the legend is sufficiently explanatory would help readers better understand the study's timeline and interventions.

Below, I provide a detailed assessment of the revised manuscript, taking into account the PLOS One Guidelines for Reviewers:

A) General Comments and Context

The manuscript follows the IMRaD structure; however, the presentation of the study design and its classification is not fully aligned with best practices for methodological description. This is particularly relevant in distinguishing between observational studies and clinical trials (or comparative experimental studies).

In my opinion, this is a comparative experimental study, as it involves a planned intervention applied to a group of participants, with measurements of physiological and metabolic outcomes. However, the manuscript does not clearly state, at the beginning of the Materials and Methods section, the exact type of study being conducted. I recommend reorganizing the Study Design subtopic as the first item in Materials and Methods, clarifying the type of study.

B) Possible Need to Follow CONSORT

Based on the study elements presented, there are indications that this could be classified as a clinical trial (even if a pilot or small-scale study). If so, adherence to the Consolidated Standards of Reporting Trials (CONSORT) is recommended to standardize the reporting of randomized clinical trials.

If this study is not strictly a clinical trial but still a controlled experimental study, the text should, at the very least, align with CONSORT’s principles of clarity and transparency, clearly describing allocation procedures, sequence randomization, and potential losses.

Additionally, if session order (e.g., normoxic vs. hypoxic) was randomized and an intervention was applied, it is necessary to partially adhere to the CONSORT checklist. I recommend verifying whether CONSORT compliance is required for this study and, if not, adopting at least its key aspects.

C) Figure Quality

The figures, especially Figure 1, remain in low resolution even after accessing the linked version. Furthermore, it seems that the diagram aims to illustrate the study chronology or protocol. However, if it serves as a clinical trial flowchart, it should include details such as the number of participants screened, any exclusions, and the distribution between conditions.

I recommend providing figures in high resolution and, if possible, making the legend more detailed, specifying each phase of the study and the number of participants in each stage.

D) Table Presentation

Table 1 is currently formatted descriptively but does not highlight the comparative aspect between groups or conditions. Given that the study aims to compare physiological/anthropometric parameters under different conditions, Table 1 could be reorganized to present these comparisons side by side, with the type of statistical analysis used clearly indicated.

I recommend revising Table 1 to align it more effectively with the standards of an experimental study.

I believe that partially or fully adopting CONSORT guidelines or another checklist for reporting experimental studies would greatly improve transparency and replicability. Additionally, I strongly recommend providing greater detail on randomization, blinding, washout periods, participant losses, and sample size calculations, reinforcing these aspects in the corresponding sections of the text.

Finally, I would like to emphasize that the topic of this article is highly relevant and presents interesting findings, particularly in the context of applications for non-athletic populations, rehabilitation, and comparative physiology studies.

**Do you want your identity to be public for this peer review?** For information about this choice, including consent withdrawal, please see our Privacy Policy

Reviewer #1: **Yes: **

Reviewer #2: No

---

## [Author Response · Author response to Decision Letter 2]

3 May 2025

Dear Reviewers,

We would like to thank you once again for the time and attention dedicated to evaluating our work.

We truly appreciate the constructive feedback, which has helped us to refine our analyses and explore aspects of the study that we had not fully considered, ultimately enhancing the clarity and the quality of our manuscript.

In the revised version, we have carefully addressed each of your comments and incorporated the necessary changes. A detailed, point-by-point response outlining how each issue has been addressed is provided in the file titled “Response to Reviewers”.

We hope that our revisions have fully addressed all your points raised.

Sincerely,

Flaminia Cesare Marincola

---

## [Editor Report · Decision Letter 2]

Metabolic response to an acute bout of mild dynamic exercise performed under normobaric moderate hypoxia: a NMR-based metabolomics study

PONE-D-24-21080R2

Dear Dr. Marincola,

We’re pleased to inform you that your manuscript has been judged scientifically suitable for publication and will be formally accepted for publication once it meets all outstanding technical requirements.

Kind regards,

Domingo Jesús Ramos-Campo, Ph.D

Academic Editor

PLOS ONE

Additional Editor Comments (optional):

The article is ready to be published
---

## [Editor Report · Acceptance letter]

PONE-D-24-21080R2

PLOS ONE

Dear Dr. Cesare Marincola,

I'm pleased to inform you that your manuscript has been deemed suitable for publication in PLOS ONE. Congratulations! Your manuscript is now being handed over to our production team.

Kind regards,

on behalf of

Dr. Domingo Jesús Ramos-Campo

Academic Editor

PLOS ONE